# Genetic Alterations in the INK4a/ARF Locus: Effects on Melanoma Development and Progression

**DOI:** 10.3390/biom10101447

**Published:** 2020-10-15

**Authors:** Zizhen Ming, Su Yin Lim, Helen Rizos

**Affiliations:** 1Faculty of Medicine, Health and Human Sciences, Macquarie University, Sydney, NSW 2109, Australia; jen.ming@mq.edu.au (Z.M.); esther.lim@mq.edu.au (S.Y.L.); 2Melanoma Institute Australia, Sydney, NSW 2065, Australia

**Keywords:** p16^INK4a^, p14^ARF^, CDKN2A, INK4a/ARF, melanoma, tumor suppressor proteins

## Abstract

Genetic alterations in the *INK4a/ARF* (or *CDKN2A*) locus have been reported in many cancer types, including melanoma; head and neck squamous cell carcinomas; lung, breast, and pancreatic cancers. In melanoma, loss of function CDKN2A alterations have been identified in approximately 50% of primary melanomas, in over 75% of metastatic melanomas, and in the germline of 40% of families with a predisposition to cutaneous melanoma. The CDKN2A locus encodes two critical tumor suppressor proteins, the cyclin-dependent kinase inhibitor p16^INK4a^ and the p53 regulator p14^ARF^. The majority of CDKN2A alterations in melanoma selectively target p16^INK4a^ or affect the coding sequence of both p16^INK4a^ and p14^ARF^. There is also a subset of less common somatic and germline INK4a/ARF alterations that affect p14^ARF^, while not altering the syntenic p16^INK4a^ coding regions. In this review, we describe the frequency and types of somatic alterations affecting the CDKN2A locus in melanoma and germline CDKN2A alterations in familial melanoma, and their functional consequences in melanoma development. We discuss the clinical implications of CDKN2A inactivating alterations and their influence on treatment response and resistance.

## 1. Introduction

Melanoma is an aggressive and highly metastatic form of skin cancer that causes nearly 60,000 deaths globally each year [1]. Melanoma originates from neural-crest-derived pigment-producing cells known as melanocytes. These specialized, dendritic cells are predominantly located in the basal layer of the epidermis, and are also found in the vascular uvea of the eye and in mucosal membranes [2]. The transformation of melanocytes into melanoma, termed melanomagenesis, involves the sequential selection of genetic and epigenetic alterations that promote proliferation, invasion, and immune escape [3,4].

Cutaneous melanoma has the highest median coding mutation rate of any cancer type (14.4 coding mutations per megabase) and this reflects the high proportion of ultraviolet (UV)-induced C>T substitutions that occur at pyrimidines [5,6]. The high mutation burden in melanoma is also associated with a high rate of silent passenger mutations and this is consistent with the 2:1 ratio of non-synonymous to synonymous mutations in melanoma [7]. The systematic evaluation of over 500 melanoma genomes in the last decade has identified a series of frequently and significantly altered oncogenes and tumor suppressor genes, including BRAF, NRAS, RAC1, NF1, CDKN2A, PTEN, and ARID2 [7,8,9].

The CDKN2A locus is located on chromosome band 9p21 and is also referred to as the INK4a/ARF locus. This is one of the most commonly altered sequences in cancer and is mutated, deleted, or methylated in 40–70% of sporadic melanomas [8]. Germline alterations in the CDKN2A locus have also been identified in approximately 40% of high-risk melanoma families with three or more melanoma cases [10]. The frequent alteration of CDKN2A reflects its capacity to encode two distinct tumor suppressor proteins, p16^INK4a^ and p14^ARF^, which are translated in alternate reading frames, from alternatively spliced transcripts with independent start sites and unique first exons (exon 1α for p16^INK4a^ and exon 1ß for p14^ARF^) [11]. p16^INK4a^ forms binary complexes with the cyclin-dependent kinases 4 and 6 (CDK4/6) to inhibit cyclin D-CDK4/6-mediated phosphorylation of the retinoblastoma protein and, thus, prevents G1 to S phase cell cycle transition [12,13,14]. The functions of p14^ARF^ are more complex, but it plays a central role in the stabilization and activation of p53 via the inhibition of the major p53 negative regulator MDM2 [15,16,17,18]. Despite sharing largely overlapping DNA sequences, the functional impact of CDKN2A alterations is complex and can be difficult to predict.

In this review, we examine genetic alterations affecting the CDKN2A locus in melanoma, the functional impact of altered CDKN2A, and the contribution of this locus to melanoma development and progression. We also discuss the impact of CDKN2A on the response of melanoma patients to current therapies.

## 2. Somatic CDKN2A Alterations in Melanoma

Inactivation of the CDKN2A locus has been detected in approximately 50% of primary melanomas and over 75% of metastatic melanomas [7,8,9]. The majority of CDKN2A alterations in melanoma selectively target p16^INK4a^ or affect the coding sequences of both p16^INK4a^ and p14^ARF^. Somatic CDKN2A alterations that solely affect p14^ARF^ without compromising the syntenic p16^INK4a^ coding region are rare. For example, analysis of the mutation spectrum of primary and metastatic cutaneous melanoma in The Cancer Genome Atlas (TCGA) (n = 363, data derived through the Memorial Sloan Kettering Cancer Center cBioPortal for Cancer Genomics [19,20] identified CDKN2A genetic alterations in approximately 45% (162/363) of melanoma cases; these included missense mutations (21/363, 5.8%), truncating mutations (34/363, 9.4%), in-frame deletions (2/363, 0.6%), amplifications (1/363, 0.3%), and homozygous deletions (112/363, 30.8%). Almost 43% of the missense, truncating, and in-frame mutations specifically affected p16^INK4a^, 57% affected both p14^ARF^ and p16^INK4a^, and mutations that altered p14^ARF^ alone were not observed. It is worth noting that alterations that appear to specifically alter one of the CDKN2A-encoded proteins may impact the co-ordinated regulation of p14^ARF^ and/or p16^INK4a^ [21]. These mutation frequencies are consistent with previous reports [22,23] and confirm that homozygous CDKN2A deletions are the most common alterations affecting this locus in melanoma (reviewed in reference [24]).

Promoter hypermethylation of CDKN2A leading to transcription silencing and loss of protein expression increases during melanoma progression and has been identified in approximately 19–60% of vertical-growth-phase melanomas, 40% of radial-growth-phase melanomas, and 25–33% of melanoma metastases [25,26,27,28]. Approximately 20–30% of melanomas show loss of p16^INK4a^ due to promoter hypermethylation [23,27], and although p14^ARF^ gene methylation is less studied, one study found that the p14^ARF^ promoter was hypermethylated in 34/60 (57%) melanoma metastases. In this study, only 16/60 (27%) metastases displayed p16^INK4a^ promoter methylation and seven of these tumors showed concurrent methylation on both the p16^INK4a^ and p14^ARF^ promoters [23]. Intriguingly, the epigenetic modifications affecting p14^ARF^ and p16^INK4a^ may vary in melanoma, with 5’ CpG promoter hypermethylation reported to be predominant in p16^INK4a^ gene inactivation, whereas histone hypoacetylation is more commonly associated with p14^ARF^ gene silencing [29]. In the TCGA cohort, methylation of three CpG islands in the p16^INK4a^ exon 1α [cg12840719, cg13601799, cg04026675] and five CpG islands in p14^ARF^ exon 1β [cg03079681, cg07562918, cg00718440, cg10848754, cg14430974] were analyzed. Although methylation levels (ß-value) were highly variable across the CDKN2A locus, the three p16^INK4a^ CpG islands appeared to be the most highly methylated in the sequence (Figure 1).

Histone methylation also regulates CDKN2A transcription. The trimethylation of histone H3 at lysine 27 (H3K27me3) along the CDKN2A locus is induced by EZH2, the catalytic subunit of the polycomb repressive complex (PRC) 2. The co-operative binding of PRC1 (including the BMI1 subunit) and the long noncoding RNA ANRIL to H3K27me3 compacts the CDKN2A genomic locus and represses transcription [25,30,31,32], in particular the transcription of p16^INK4a^ [33]. In line with these data, the levels of H3K27me3 increase from primary to metastatic melanoma, and high H3K27me3 is an independent poor prognostic marker in melanoma [34,35].

Mutations in the BRAF, NRAS, and NF1 genes are the predominant drivers in cutaneous melanoma, and CDKN2A genetic alterations are distributed evenly amongst these genetic melanoma subtypes, with CDKN2A genetic alterations detected in 48.9% (92/188) of BRAF-mutant, 44.7% (51/114) of NRAS-mutant, and 50% (33/66) of NF1-mutant cutaneous melanomas (Table 1) [8]. In contrast, CDKN2A promoter hypermethylation occurs most frequently in NRAS-mutant melanoma (87.7%) and is less common in melanoma with BRAF mutations (67.0% with CDKN2A promoter methylation) and in melanomas with NF1 mutations (77.3% showing CDKN2A promoter methylation) (Table 1). Analysis of the TCGA cutaneous melanoma dataset also showed that methylation of the p16^INK4a^ probe cg12840719 was higher in NRAS-mutant melanoma, compared to BRAF-mutant melanoma, although this was not statistically significant (Figure 1C, Table 1). Interestingly CDKN2A methylation was rarely detected in sporadic primary melanoma and has not been identified in melanocytic nevi [36,37]. Altogether, CDKN2A alterations (including mutation, deletion, and promoter hypermethylation) appear to be differentially distributed across the cutaneous melanoma genomic subtypes, and have been identified in 58% of BRAF-mutant, 72% of NRAS-mutant, and 71% of NF1-mutant melanoma but only in 37% of triple wild-type melanomas [8].

Further analysis of TCGA skin cutaneous melanoma dataset revealed that the frequency of CDKN2A genetic alterations was not influenced by gender; 63/135 (47%) females and 98/228 (43%) males showed melanoma-associated CDKN2A genetic changes, and this is in keeping with a previous report [38]. Melanoma patients diagnosed at a younger age tend to have a higher frequency of CDKN2A somatic alterations, however, with 72% of patients aged 30 years or younger displaying melanoma-associated CDKN2A alterations, and this diminishes with increasing age. In particular, 50% of melanoma patients aged 31–50, 41% of patients aged 51–70, and 36% of patients aged > 71 had somatic CDKN2A genetic changes [8]. Further, CDKN2A mutation status is not associated with overall survival in melanoma; melanoma patients with somatic CDKN2A alterations had a median overall survival of 112 months compared to a median overall survival of 79 months in patients with wild-type CDKN2A [8] (Figure 2).

CDKN2A is also highly mutated in rarer subtypes of cutaneous melanoma and in non-cutaneous melanoma. For instance, CDKN2A mutations are present in 20% (4 out of 20 cases; 3 truncating mutations and one missense mutation) of desmoplastic melanoma, and these alterations predominantly affected either p16^INK4a^ alone or p16^INK4a^ along with p14^ARF^ [39]. CDKN2A mutations are less common in acral melanoma, with only 9–18% of acral melanomas showing somatic CDKN2A alterations, and these alterations were predominantly homozygous deletions [40,41]. CDKN2A mutation rates were similarly low in uveal melanoma, with only one missense mutation (V28G within exon 1α) detected in the 80 uveal melanomas included in TCGA uveal melanoma cohort. However, approximately 32% of primary uveal melanomas and 50% of uveal melanoma cell lines show hypermethylation of the CDKN2A promoter, and this promoter methylation predominantly affected p16^INK4a^ expression [27,28,29].

## 3. Germline CDKN2A Alterations in Familial Melanoma

Loss of function alterations affecting the CDKN2A locus have been identified in the germline of multiple-case melanoma kindreds around the world [42,43]. These CDKN2A germline mutations are associated with a 65-fold increase in the risk of melanoma development [44]. Individuals in families with CDKN2A alterations commonly have higher numbers of atypical melanocytic nevi [45]. The systematic analysis of 80 melanoma-prone families identified 37 distinct mutations affecting the CDKN2A locus. Half of these mutations were in exon 1α (these will not affect the p14^ARF^ coding sequence) and half were in exon 2. Whereas all exon 2 mutations altered the p16^INK4a^ protein, only 14/20 of these exon 2 mutations also impacted the p14^ARF^ protein sequence [45].

As is the case with CDKN2A somatic alterations, exon 1β-specific genetic alterations affecting p14^ARF^ alone are less frequent than mutations targeting the p16^INK4a^ coding regions (Table 2). Germline inactivation of p14^ARF^ is mostly due to whole gene deletions, insertions, or splice mutations [46,47,48]. For instance, a germline deletion of p14^ARF^ exon 1β coding sequence and a germline mutation (Gly16Asp) in exon 1β were identified in families with melanoma-neural system tumor syndrome [47,49]; a frameshift mutation (16 base pair insertion) was detected in a Spanish female who developed multiple primary melanomas at a young age [50]; a splice mutation resulting in haploinsufficiency was associated with melanoma in a single family [51]; and a mutation coding for the missense Arg54His mutation in p14^ARF^ was found in an Italian melanoma family [52]. Large germline CDKN2A deletions encompassing exon 1α, 2, and/or 3 which affect both p14^ARF^ and p16^INK4a^ have also been identified in melanoma-prone kindreds [53,54], although these are uncommon and account for only 2% of germline CDKN2A alterations [55]. A germline splice site mutation removing exon 2 of CDKN2A was identified in a family with melanoma and multiple dysplastic nevi [56].

The penetrance of germline CDKN2A mutations for melanoma also varies according to geographical locations, with penetrance increasing with higher baseline melanoma incidence rates. For instance, lifetime penetrance of CDKN2A mutations was 0.58 in Europe, 0.76 in the United States, and 0.91 in Australia [38]. The median age of melanoma diagnosis was also younger in Australian melanoma-prone families compared to European families [10]. A GenoMEL study that screened for CDKN2A mutations in Australian, European, and North American families reported increased mutation frequency in families with melanoma and pancreatic cancer [10]. A prospective study examining cancer diagnoses in Swedish carriers of the Arg112dup alteration in CDKN2A concluded that mutation carriers had a significantly elevated risk of developing pancreatic, lung, head and neck, and gastro-oesophageal carcinomas [57]. Based on five clinical features (number of family members with melanoma, number of members with multiple primary melanomas, median age at diagnosis, presence of pancreatic cancer, and presence of upper airway cancer), Potjer et al. developed the CDKN2A Mutation (CM) score to predict CDKN2A germline mutation status among melanoma prone families; a CM score > 35 out of 49 possible points was associated with a > 90% probability of a melanoma-prone family sharing a CDKN2A mutation [58]. Importantly, CDKN2A mutation carriers have been reported to be at increased risk of developing other cancers, including breast, lung, and non-melanoma skin cancers [57,59,60,61,62]. These additional cancer risks are not consistently observed, however, and this may indicate that the risk of other cancer types reflects the specific germline CDKN2A mutation. For instance, whereas 25–36% of melanoma-prone families with Arg24Pro, c.34G>T, or Gly101Trp had pancreatic cancer, none of the 30 families with Met53Ile, c.IVS2-104A>G, or c.32_33ins9-32 developed pancreatic cancer [10]. Finally, a CDKN2A germline deletion of the p14^ARF^-specific exon 1ß was associated with excess risk for melanoma, astrocytoma, neurofibromas, and schwannomas [47].

Modifier genes for melanoma kindreds carrying CDKN2A mutations have also been reported. In an early study on Dutch melanoma families, the melanocortin-1 receptor (MC1R) variant Arg151Cys increased the risk of melanoma in carriers of a p16^INK4a^-inactivating deletion (known as p16-Leiden). This increased risk of melanoma was not wholly attributed to the fair skin type associated with this MC1R variant [63]. A later study confirmed that four frequent MC1R variants (Val60Leu, Val92Met, Arg151Cys, Arg160Tro) were associated with an increased melanoma risk in CDKN2A mutation carriers [64]. Polymorphisms in genes involved in DNA repair (POLN, PRKDC), immune regulation (IL9), and apoptosis (BCL7A, BCL2L1) have also been associated with increased melanoma risk, and in some instances, these polymorphisms (IL9 and BCL7A) have stronger risks in CDKN2A-positive families [65,66].

## 4. Functional Consequences of Melanoma-Associated CDKN2A Alterations

### 4.1. p16^INK4a^ Mutations

Melanoma-associated CDKN2A missense mutations commonly diminish the capacity of p16^INK4a^ to bind and inhibit CDK4/6 [67]. For example, Ranade et al. described germline CDKN2A substitutions that impaired the ability of p16^INK4a^ to inhibit the catalytic activity of cyclin D1/CDK4 and cyclin D1/CDK6 complexes [14]. The Met53Ile and Arg24Pro germline mutants of p16^INK4a^ have diminished capacity to bind CDK4 compared to wild-type p16^INK4a^ [68]. Although, it is worth noting that the CDK4-binding affinity of the Arg24Pro mutation is controversial [69]. The somatic missense p16^INK4a^ mutation (Pro48Leu) decreased the ability of the protein to bind and inhibit CDK6 kinase activity, thus, failing to arrest melanoma cell growth [70]. Melanoma cell lines with a Pro81Leu missense mutation in p16^INK4a^ also showed defective binding ability to CDK4, and these cells had more aggressive cell growth compared to the wild-type cells [71].

There is some evidence that CDKN2A mutations do not only impact the binding affinity of p16^INK4a^ for CDK4 or CDK6. Indeed, several melanoma-associated mutations (e.g., N-terminal 24bp p16^INK4a^ duplication, Arg24Pro, Leu117Pro) retained CDK4 and/or CDK6 binding activity even though they displayed diminished cell cycle inhibitory activity, suggesting that other p16^INK4a^ binding interactions may be important in melanoma susceptibility [69,72]. p16^INK4a^ mutants may also dysregulate the oxidative stress response independent of cell cycle regulation. Elevated levels of intracellular reactive oxygen species and oxidative DNA damage were observed in p16^INK4a^-deficient melanocytes and cancer cells, likely due to altered p38 MAPK signaling [73]. Various p16^INK4a^ mutants, including the melanoma-associated germline mutations Ala36Pro and Ala57Val, were also associated with impaired oxidative regulatory functions [74], and intracellular oxidative dysregulation in melanocytes can lead to genetic damage that contributes to increased melanoma susceptibility [73].

Inactivation of p16^INK4a^ has been shown to contribute to the failure of senescence and progression from normal melanocytes to malignant melanoma via benign nevi, dysplastic nevi, radial growth phase, and vertical growth phase stages [75,76]. Loss of p16^INK4a^ as a single event is not sufficient to induce melanomagenesis but does predispose one to melanoma development, especially in the presence of other driver mutations. For instance, loss of p16^INK4a^ cooperates with BRAF^V600E^ oncogenic mutation to promote melanoma progression in genetic mouse models [77]. More recently, Zeng et al. reported that bi-allelic deletion of p16^INK4a^ induced hyper-motile and invasive melanocyte behavior via the transcriptional activation of the lineage-restricted transcription factor BRN2 [78].

### 4.2. p14^ARF^ Mutations

The functional impact of exon 1β variants which alter only p14^ARF^ has been less well studied. Specific germline deletions of exon 1β resulting in loss of p14^ARF^ have been identified in melanoma cell lines and in a family with melanoma-neural system tumor syndrome [47,49]. A 16 base pair insertion (60ins16) caused by a duplication of a CG-rich region within exon 1β was detected in a Spanish female who had multiple primary melanomas. This mutant p14^ARF^ retained the N-terminal nucleolar localization sequence but was restricted to the cytoplasm, failed to stabilize p53, and did not induce cell cycle arrest in p53-expressing melanoma cells [50]. In contrast, another p14^ARF^ variant (Gly16Asp) in exon 1β retained its ability to bind HDM2 and stabilize p53 [49]. A splice site mutation in exon 1β was shown to cause p14^ARF^ haploinsufficiency and was associated with melanoma susceptibility [79]. There have been several studies investigating the role of CDKN2A exon 2 mutations that impact the p14^ARF^ coding sequence. In one study, 3/7 p14^ARF^ mutations, encoded by CDKN2A exon 2 mutations, altered the subcellular distribution of p14^ARF^ and diminished its ability to stabilize p53 [50].

## 5. Clinical Implications of CDKN2A: Impact on Response and Resistance to Current Treatments in Melanoma

The restoration of p14^ARF^ and/or p16^INK4a^ functions has not yet been possible, and most therapeutic strategies involve modulating downstream cell cycle regulators or pathways to overcome the loss of CDKN2A-encoded functions.

### 5.1. MDM2 Inhibitors

To overcome p14^ARF^ loss, small molecule inhibitors targeting MDM2 activity or the MDM2–p53 interaction have been developed. These inhibitors have shown promising anti-tumor effects in the preclinical setting. The small molecules nutlins (nutlin-1, nutlin-2, and nutlin-3) sterically disrupt the interaction between MDM2 and p53, resulting in p53 accumulation and activation [80]. Nutlin-3 in particular has been shown to inhibit melanoma growth and induce apoptosis in patient-derived xenograft models [81]. Similarly, the MDM2 inhibitor KRT-232 inhibited tumor growth in xenografts derived from 15 melanoma patients, when used alone or in combination with BRAF and/or MEK inhibitors [82]. Importantly, although MDM2 inhibitors will only benefit melanoma patients with p53 wild-type tumors, melanoma often retains expression of wild-type p53 [81].

### 5.2. CDK Inhibitors

To circumvent p16^INK4a^ loss, CDK inhibitors have been developed and tested with variable success. The first-generation CDK inhibitor, flavopiridol, has broad range activity against CDK1, CDK2, CDK4, and CDK7 and induced cell cycle arrest in preclinical melanoma models but failed to generate any significant clinical activity in a phase II trial of metastatic melanoma patients (NCT00005971 [83]). Second generation CDK-specific inhibitors such as ribociclib (LEE011) and abemaciclib selectively target CDK4 and CDK6, and these have shown more promising results, especially when combined with MAPK inhibitors. For instance, ribociclib in combination with binimetinib (MEK inhibitor) enhanced tumor regression in NRAS^Q61K^-mutant melanoma xenograft models compared to treatment with either ribociclib or binimetinib alone [84]. Ribociclib also demonstrated synergistic effects in combination with encorafenib (BRAF inhibitor) in BRAF^V600E^-mutant melanoma models. Likewise, treatment with the selective CDK4/6 inhibitor abemaciclib inhibited tumor growth and delayed tumor recurrences in melanoma xenograft mouse models. Importantly, abemaciclib caused tumor regression in vemurafenib (BRAF inhibitor)-resistant tumors, suggesting that CDK4/6 inhibitors may be a viable therapeutic option for melanoma patients who progressed on BRAF/MEK inhibitors [85]. The combination of ribociclib and MDM2 inhibition also enhanced tumor regression and overcame resistance to CDK4/6 inhibitors in a melanoma xenograft model [86].

Selective CDK4/6 inhibitors have since been evaluated in clinical trials. The combination of CDK4 and MEK inhibitors (ribociclib and binimetinib) was tested in a phase Ib/II trial with advanced NRAS-mutant melanoma, and 60–70% of patients experienced clinical benefit (RECISR CR, PR, and SD) [87,88]. The combination of ribociclib and the BRAF inhibitor encorafenib was also evaluated in 18 patients with advanced BRAF-mutant melanoma, with more than half of patients showing clinical benefit (PR/SD) [89]. A triple combination of ribociclib, binimetinib, and encorafenib was evaluated in a phase Ib/II study of 21 patients with BRAF-mutant melanoma, and although increased toxicity was observed with this combination, clinical response was noted in over half of the patients [90].

Loss of CDKN2A has been shown to predict response to CDK4/6 inhibitors in melanoma, glioblastoma, ovarian, and rhabdoid tumor cells [91,92,93]. The activity of CDK4/6 inhibitors was also restricted to melanoma cells that retained expression of the retinoblastoma protein (the downstream effector of CDK4 and CDK6). The presence of an activating Arg24Cys CDK4 mutation, which abolishes the ability of CDK4 to bind to p16^INK4^ [94] was also associated with melanoma cell sensitivity to CDK4/6 inhibition [95]. These data suggest that activation of the CDK4/6 pathway via loss of p16^INK4a^ or CDK4 activation and the retention of the retinoblastoma protein are key determinants of sensitivity to CDK4/6 inhibition.

### 5.3. Epigenetic Modulators

Considering that CDKN2A methylation can lead to p14^ARF^ and p16^INK4a^ loss, epigenetic reactivation of CDKN2A has also been attempted with inhibitors of DNA methyltransferase (DNMT), histone deacetylase (HDAC), histone methyltransferase, and histone acetyltransferase. These inhibitors have been shown to induce p14^ARF^ and p16^INK4a^ expression in cancer cell lines and preclinical models (reviewed in reference [96]). In melanoma, treatment of melanoma cell lines with 5-aza-2-deoxycytidine, a DNMT inhibitor, and vorinostat, an HDAC inhibitor, restored p14^ARF^ and p16^INK4a^ function, and this led to reduced cell proliferation, migration, and invasion [29]. However, given that these epigenetic modulators have promiscuous effects, it is difficult to attribute the consequent melanoma control on modulation of p14^ARF^ and p16^INK4a^ function alone.

### 5.4. BRAF/MEK Inhibitors

The BRAF^V600^ inhibitors dabrafenib and vemurafenib, in combination with MEK inhibitors trametinib or selumetinib, have now become standard of care in BRAF^V600^-mutant melanoma. The combination of dabrafenib and trametinib produced response rates of 64%, and progression-free survival and overall survival rates of 13% and 28% at 5 years, respectively, in patients with advanced BRAF^V600^-mutant melanoma [97], superior to single-agent treatment. However, despite improved response and progression-free survival rates, melanoma patients treated with these selective kinase inhibitors quickly develop resistance and progress within one year.

Despite the high frequency of CDKN2A alterations in melanomas, the impact of CDKN2A mutations on patient responses to BRAF/MEK inhibitors is not well established. For instance, in melanoma cell studies, the presence of p16^INK4a^-resistant CDK4 mutations (including the melanoma-associated germline CDK4 Arg24Cys mutation) did not alter cell sensitivity to BRAF inhibitors. Conversely, the overexpression of cyclin D1 was associated with BRAF inhibitor resistance and resistance was enhanced when cyclin D1 overexpression was combined with the CDK4 Arg24Cys mutation [98]. Recurrent CDKN2A loss has been implicated in BRAF inhibitor resistance [99,100], although CDKN2A alterations have been found to be pre-existing in responding patients, and CDKN2A alterations commonly co-occur with other mechanisms of BRAF inhibitor resistance (i.e., PTEN loss, N-RAS mutations) [101]. Finally, although reduced CDKN2A copy number at baseline has been associated with poor BRAF inhibitor responses in melanoma [102], the genetic loss of CDKN2A is also a poor prognostic marker in melanoma [103].

It is important to mention that 15–40% of mucosal and acral melanomas show activating mutations or amplification of the receptor tyrosine kinase KIT, and the kinase inhibitor imatinib has shown efficacy in KIT-mutant melanoma with an overall response rate of 54% [104]. Imatinib is also used commonly in the treatment of BCR–ABL chronic myelogenous leukemia but has not been as successful in BCR–ABL positive acute lymphoblastic leukemia showing deletion in the CDKN2A gene, suggesting that expression of p14^ARF^ and/or p16^INK4a^ may sensitize cancer cells to imatinib treatment [105]. Thus, it is tempting to speculate that CDKN2A inactivation in melanoma may analogously diminish sensitivity to imatinib in melanoma.

### 5.5. Immune Checkpoint Inhibitors

The immune checkpoint inhibitors targeting the inhibitory receptors cytotoxic T lymphocyte-associated antigen 4 (CTLA-4) and programmed death-1 (PD-1) have significantly improved survival of patients with advanced and high-risk stage III melanoma. The CTLA-4 inhibitor ipilimumab generates a response rate of around 20% in melanoma patients, and a small proportion of patients remain disease-free past 10 years [106]. Response rates are higher with PD-1 inhibitors (up to 45%) and the combination of CTLA-4 and PD-1 inhibitors further enhances the response rate to 60% [107,108,109,110].

Immune checkpoint inhibitors show most activity in immunogenic cancers, and in tumors showing IFNγ transcriptome signatures and evidence of infiltrating T cells [111]. In this context, knockout of the CDKN2A gene in mice resulted in increased inflammatory cytokine expression in the skin following chronic UVB irradiation. Additionally, more myeloid cells were identified in the CDKN2A knockout mice [112]. Interestingly, chromosomal 9p losses encompassing CDKN2A can also affect the JAK2 gene (JAK2 is located on chromosome band 9p24.1 [113]). JAK2 is a critical transcription factor in IFNγ signaling, and the loss of JAK2 is associated with PD-1 inhibitor resistance [114]. Indeed, 75% of melanoma tumors carry concurrent loss of the JAK2 and CDKN2A alleles [115]. Hence, loss of CDKN2A may increase inflammatory responses, which may augment response to immune checkpoint blockade, but also confer susceptibility to immunotherapy resistance through IFNγ suppression. Given the complexity of the immune response and the heterogeneity of immune cell subsets, it is unclear if and how p14^ARF^ and/or p16^INK4a^ regulate melanoma response to immunotherapy.

CDKN2A mutations were not significantly associated with clinical outcomes such as median time to progression, overall survival, and disease control rate in a cohort of 102 cutaneous melanoma patients treated with immune checkpoint inhibitors. However, this study did report a trend towards improved time to progression and disease control rate in patients with CDKN2A mutations [116]. Similarly, melanoma patients with CDKN2A germline mutations also showed improved response to immune checkpoint blockade; approximately 58% of carriers responded to therapy, with 32% showing complete response [117], suggesting that CDKN2A mutation may be associated with better immunotherapy response rates. Although the mechanism for improved immunotherapy responsiveness in CDKN2A mutation carriers remains unclear, melanomas with somatic CDKN2A mutations have an increased mutational burden, and this may result in more neoantigens and stronger immune responses [117].

### 5.6. Chemotherapy

Chemotherapy is still used as salvage treatment for melanoma patients, especially those with BRAF wild-type disease, and in patients who have failed molecular targeted and/or immunotherapy [118]. Chemotherapy agents such as dacarbazine and temozolomide show low response rates of 12–13% and median overall survival of only 6–8 months [119], and although partial response rates for these agents can reach 15–28%, less than 2% of patients will have durable responses (reviewed in [120]).

Expression of p14^ARF^ has been shown to enhance chemosensitivity. For example, p14^ARF^ accumulation induced potent cell cycle arrest in a p53-dependent manner. On its own, p14^ARF^ did not induce apoptosis, but rather sensitized cells to apoptosis in the presence of camptothecin and adriamycin, inhibitors of topoisomerase I and II, in osteosarcoma, colorectal, melanoma, and fibroblast cell lines [121]. This effect is also observed in osteosarcoma cell lines in response to cisplatin-induced apoptosis, however, effects were independent of p53 [122], suggesting a distinct regulatory mechanism that may be treatment dependent.

Similar to p14^ARF^, ectopic expression of p16^INK4a^ in glioma cell lines also sensitized cells to the chemotherapy drug vincristine [123]. In melanoma cells, CDKN2A expression was associated with better response to chemotherapy in the form of melphalan or actinomycin-D, and enforced accumulation of p16^INK4a^ induced cell death by augmenting response to these cytotoxic drugs [124].

## 6. Conclusions

The CDKN2A locus is the most common melanoma-dominant predisposition gene and somatic alterations encompassing this genetic sequence occur early in the development of melanoma. Many CDKN2A genetic and epigenetic changes impact both the p16^INK4a^ and p14^ARF^ protein products encoded by this locus, and although early studies confirmed the major contribution of p16^INK4a^ in CDKN2A-associated melanoma, there is now significant evidence that p14^ARF^ plays an important and additional role in melanomagenesis. CDKN2A loss is associated with histological features predictive of poor prognosis in melanoma and also correlates with diminished patient response to treatment, with loss of CDKN2A associated with poor response to BRAF/MEK inhibitors and chemotherapy but potentially improved responses to immune checkpoint inhibitors. The loss of the CDKN2A sequence also co-operates with the BRAF and NRAS oncogenes to promote melanoma development. Thus, there is renewed interest in restoring the functional loss of p16^INK4a^ and p14^ARF^ in melanoma, and the frequent loss of this locus in melanoma may provide unique therapeutic opportunities, as the downstream targets retinoblastoma protein and p53 are often retained. In recent work, the combination of CDK4 and MDM2 inhibitors demonstrated significant preclinical activity, and this combination was effective in melanoma models with genetic loss of CDKN2A. It remains to be determined whether combination therapies that functionally restore CDKN2A will be effective as salvage therapies. The CDKN2A locus may ultimately help stratify patients for optimal treatment and provide therapeutic options for patients who fail standard of care MAPK inhibitor-based and/or PD1-inhibitor-based therapies.

## Figures and Tables

**Figure 1 biomolecules-10-01447-f001:**
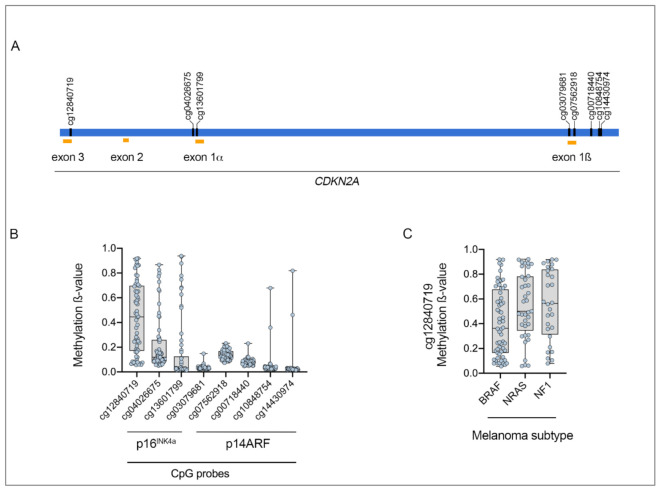
CDKN2A methylation (HumanMethylation450 arrays) ß-values (β-values range from 0, unmethylated, to 1, fully methylated). (**A**) Location of methylation probes and exons across the CDKN2A locus. (**B**) Methylation status of probes located across the CDKN2A locus. (**C**) Methylation of the cg12840719 probe according to the melanoma genomic subtypes. Data are derived from The Cancer Genome Atlas (TCGA) Skin Cutaneous Melanoma (SKCM) dataset, and only include 62 melanoma tumors without CDKN2A genomic deletions.

**Figure 2 biomolecules-10-01447-f002:**
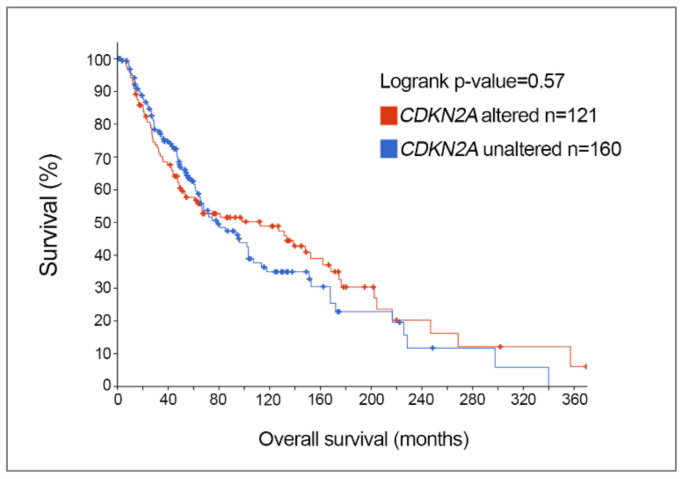
Kaplan–Meier overall survival curve of patients from TCGA skin cutaneous melanoma cohort with altered vs. unaltered (wild-type) CDKN2A. Median overall survival for patients with CDKN2A alterations is 112 months, compared to a median overall survival of 79 months for melanoma patients with unaltered CDKN2A (*p* = 0.57, logrank test).

**Table 1 biomolecules-10-01447-t001:** Frequency and type of CDKN2A alterations in the four cutaneous melanoma genomic subtypes, designated BRAF, RAS, NF1, and triple wild-type.

Molecular Subtype	BRAF (n = 188)	NRAS (n = 114)	NF1 (n = 66)	Triple Wild-Type (n = 45)
**Mutations in p16^INK4a^ only**	7.4% (14/188)	6.1% (7/114)	9.1% (6/66)	0% (0/45)
**Mutations Affecting p16^INK4a^ and p14^ARF^**	6.9% (13/188)	7.9% (9/114)	12.1% (8/66)	4.4% (2/45)
**CDKN2A Deletions**	34.6% (65/188)	30.7% (35/114)	28.8% (19/66)	17.8% (8/45)
**CDKN2A Hyper-Methylation**	67.0% (126/188)	87.7% (100/114)	77.3% (51/66)	66.7% (30/45)

Data derived from TCGA Skin Cutaneous Melanoma dataset via cBioPortal for Cancer Genomics [19]. CDKN2A hyper-methylation defined as methylation levels ß-value > 0.2 in any of the CDKN2A probes located across the CDKN2A (see Figure 1).

**Table 2 biomolecules-10-01447-t002:** Frequency and type of germline CDKN2A alterations identified in 676 individuals with various cancers, including melanoma and pancreatic cancers.

Germline Mutations	CDKN2 ANM_000077.4 (p16^INK4a^)	CDKN2A NM_058195.3 (p14^ARF^)
**Total Number of Coding Variants**	435	179
**Number of Coding Variants**		
**Exon 1a**	140/435 (32%)	0/477 (0%)
**Exon 1b**	0/435 (0%	2/477 (0%)
**Exon 2**	295/435 (68%)	177/477 (37%)
**Exon 3**	0/435 (0%)	0/477 (0%)

Data derived from LOVD^3^, Leiden Open Variant Database.

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
