# Peer review of "Genetic Alterations in the INK4a/ARF Locus: Effects on Melanoma Development and Progression"

_biomolecules, 2020, doi:10.3390/biom10101447_

Round 1
Reviewer 1 Report
This is a comprehensive review on the subject of INK4a/ARF locus mutations and genomic, epigenomic changes in melanoma. It is highly comprehensive and well described.
A major problem is with the meaning of the recurrently used term “cutaneous melanoma”. “Cutaneous” by definition is “relating to or affecting the skin”. Yet, the authors referenced to desmoplastic melanoma and acral melanoma as non-cutaneous. They are both cutaneous. “Desmoplastic melanoma (DM) is a variant of spindle cell melanoma typically found on chronically sun-damaged skin of older individuals” and “Acral melanoma, sometimes called acral lentiginous melanoma, is a rare type of skin melanoma that forms on the palms, soles of feet, or under finger or toe nails.” The only difference between the two is exposure to the sun. It would be between if the authors “divide” their analysis to sun-exposed and sun-shielded melanomas, and consider uveal and mucosal melanoma in a different category.
Author Response
REVIEWER 1
- Cutaneous melanoma – correct referencing desmoplastic and acral melanoma
We have updated the introduction to this paragraph as shown below:
‘CDKN2A is also highly mutated in rarer subtypes of cutaneous melanoma and in non-cutaneous melanoma’
Reviewer 2 Report
This review presents a valuable collection of data on the CDKN2A gene in melanoma, is logically structures and very well written. My comments therefore should be viewed for the most part as suggestions to improve this work.
In figure 1 the CpG sites and average beta values are given. It seems to me that an illustration of the location of the CpG sites at the CDKN2A location could provide the reader more insight. Not only for promoter hypermethylation, but also for somatic mutations and germline mutations, visualizing location in a single figure or in multiple panels would clarify the description in the text and might even provide new insights. Perhaps the authors might consider adding such a figure.
The frequency of CDKN2A promoter hypermethylation as described in the text and in Table 1 to me did not seem to match. It might be an error on my side, but this should be checked.
Therapeutic efficacy of CDK4/6 inhibitors is discussed, but not in the light of presence or absence of CDKN2A alterations. If such data are available the authors could consider adding to this section.
At line 94 it is stated that BRAF, NRAS and NF1 are the most frequently mutated genes in melanoma. This is not the case; mutations in these genes are considered the three main drivers (of MAPK signalling hyperactivation).
When discussing about p14ARF mutations, the authors might consider to mention that mutations that do not affect p16 coding region might still affect transcriptional regulation of p16.
When discussing about germline variants using and referring to a central database, repository, resource such as Leiden Open Variation Database could be considered, similar to how TCGA has been used for somatic mutations.
It is impossible to be complete, but modifier genes for melanoma and sarcoma risk as well as founder mutations in CDKN2A as well as evolution of the CDKN2A-CDKN2B locus (p14, p15, p16) could be added. Also more could be added on transcriptional regulation of CDKN2A (enhancers, ANRIL, CDC6, BMI1).
Minor are in line 45 alternative splicing is mentioned; also the transcription start site differs (exon 1a and exon 1b) for p16 and p14. In line 277 induced should be induce.
Author Response
REVIEWER 2
- Illustration on CpG sites
We have included a new panel in Figure 1 showing the location of CDKN2A exons and the methylation probes shown in Figure 1B. It was not possible to include CDKN2A mutations in this Figure due to the large genomic region shown
- Frequency of CDKN2A promoter methylation in text and Table 1 did not match
We appreciate the reviewer’s comment. The frequency of CDKN2A promoter hypermethylation as described in the text was cited from previous published data, whereas the data shown in the table was based on TCGA data. We have now reconciled this information for consistency.
- Therapeutic efficacy of CDK4/6 inhibitors is discussed, but not in the light of presence or absence of CDKN2A alterations.
We have now expanded the section on CDK4/6 inhibitors and include details on predictors of response (page 10, paragraph 2)
- At line 94 it is stated that BRAF, NRAS and NF1 are the most frequently mutated genes in melanoma. This is not the case; mutations in these genes are considered the three main drivers (of MAPK signalling hyperactivation).
The sentence has been reworded to “Mutations in BRAF, NRAS and NF1 genes are the predominant drivers in cutaneous melanoma”
- When discussing about p14ARF mutations, the authors might consider to mention that mutations that do not affect p16 coding region might still affect transcriptional regulation of p16.
We have added the following sentence and reference to indicate that mutations can alter the coordinated regulation of p16 and/or p14ARF.
‘It is also worth noting that alterations that appear to specifically alter one of the CDKN2A-encoded protein may also impact the co-ordinated regulation of p14ARF and/or p16INK4a {Rizos, 1997 #912}.’
- When discussing about germline variants using and referring to a central database, repository, resource such as Leiden Open Variation Database could be considered
A new Table 2 has been included with summary of germline mutations affecting the CDKN2A locus, data has been derived from the Leiden Open Variant Database
- Additional details on modifier genes and transcription regulation
We have included a new paragraph to highlight the impact of modifier genes on CDKN2A-carrier melanoma risk. We decided not to include evolution of CDKN2A as although interesting is not essential for the topic we are reviewing. We also added a little more on the regulation of the CDKN2A locus, throughout the review, but only in relation to its association with melanoma.
- Minor are in line 45 alternative splicing is mentioned; also the transcription start site differs (exon 1a and exon 1b) for p16 and p14. In line 277 induced should be induce.
Corrections have been made